# Proliferative Potential, and Inflammatory Tumor Microenvironment in Meningioma Correlate with Neurological Function at Presentation and Anatomical Location—From Convexity to Skull Base and Spine

**DOI:** 10.3390/cancers14041033

**Published:** 2022-02-18

**Authors:** Johannes Wach, Tim Lampmann, Ági Güresir, Hartmut Vatter, Ulrich Herrlinger, Albert Becker, Silvia Cases-Cunillera, Michael Hölzel, Marieta Toma, Erdem Güresir

**Affiliations:** 1Department of Neurosurgery, University Hospital Bonn, 53127 Bonn, Germany; tim.lampmann@ukbonn.de (T.L.); agi.gueresir@ukbonn.de (Á.G.); hartmut.vatter@ukbonn.de (H.V.); erdem.gueresir@ukbonn.de (E.G.); 2Division of Clinical Neurooncology, Department of Neurology, University Hospital Bonn, 53127 Bonn, Germany; ulrich.herrlinger@ukbonn.de; 3Department of Neuropathology, University Hospital Bonn, 53127 Bonn, Germany; albert_becker@uni-bonn.de (A.B.); cases_92@hotmail.com (S.C.-C.); 4Institute of Experimental Oncology, University Hospital Bonn, 53127 Bonn, Germany; michael.hoelzel@ukbonn.de; 5Institute of Pathology, University Hospital Bonn, 53127 Bonn, Germany; marieta.toma@ukbonn.de

**Keywords:** anatomic location, macrophages, meningioma, MIB-1 index, tumor progression

## Abstract

**Simple Summary:**

The World Health Organization (WHO) classification grades meningiomas exclusively due to their histopathological features. Meningiomas are predominantly benign intracranial entities, and surgical resection represents the therapy of choice. However, risk of progression and tailored scheduling of follow-up appointments are significantly influenced by various items, such as immunohistochemistry (e.g., MIB-1 index). Emerging evidence focuses attention on the anatomic location of meningiomas, especially regarding the differentiation between skull base and non-skull base meningiomas. In the present study, we therefore investigated demographic, histopathological, and laboratory variables regarding their association with the anatomic location. We found that spinal meningiomas have a significantly lower proliferative activity, less density of macrophage infiltrates, and a longer time to tumor progression. Moreover, increased MIB-1 indices are significantly associated with location-specific baseline symptoms (e.g., convexity: seizure burden, medial skull base: decreased vision, spinal: ambulatory ability). Therefore, anatomic location might be considered as a future subclassification in the grading of the prognosis of meningiomas.

**Abstract:**

Emerging evidence emphasizes the prognostic importance of meningioma location. The present investigation evaluates whether progression-free survival (PFS), proliferative potential, World Health Organization (WHO) grades, and inflammatory burden differ between anatomical locations (skull base, non-skull base, and spinal) meningiomas. Five-hundred-forty-one patients underwent Simpson grade I or II resection for WHO grade 1 or 2 meningiomas. Univariable analysis revealed that spinal meningioma patients are significantly older, had a worse baseline Karnofsky Performance Status (KPS), higher acute-phase protein levels, lower incidence of WHO grade 2, lower mitotic counts, lower MIB-1 index, and less CD68^+^ macrophage infiltrates. Multivariable analysis identified WHO grade 2 (OR: 2.1, 95% CI: 1.1–3.7, *p* = 0.02) and cranial location (OR: 3.0, 95% CI: 1.8–4.9, *p* = 0.001) as independent predictors of diffuse CD68^+^ macrophage infiltrates. The mean PFS in cranial meningiomas was 115.9 months (95% CI: 107.5–124.3), compared to 162.2 months (95% CI: 150.5–174.0; log-rank test: *p* = 0.02) in spinal meningiomas. Multivariable Cox regression analysis revealed cranial location as an independent predictor (HR: 4.7, 95% CI: 1.0–21.3, *p* = 0.04) of shortened PFS. Increased MIB-1 indices ≥5% were significantly associated with location-specific deficits at presentation, such as decreased vision and seizure burden. Spinal meningiomas have a significantly longer PFS time and differ from the cranial meningiomas regarding MIB-1 index and density of tumor-associated macrophages.

## 1. Introduction

Meningiomas are predominantly benign tumors, which account for 36.4% of all central nervous system (CNS) neoplasms [1,2]. Gross total microsurgical resection is the initial therapy of choice for World Health Organization (WHO) grade 1 and 2 meningiomas, which account for 97–99% of all meningiomas [3,4]. Moreover, MR-imaging features, blood-based markers, adjuvant radiotherapy, and neuropathological characteristics are intensively discussed regarding the prediction of meningioma progression [5,6,7,8].

Immunohistochemical methods such as the Molecular Immunology Borstel (MIB-1)-1 index are increasingly debated as predictors of progression-free survival in meningiomas [9,10,11,12]. MIB-1 index is an immunohistochemical method to detect nuclear structures that are exclusively visible in cells during the proliferation and facilitates the determination of the growth fraction within a neoplastic tissue sample. Antibodies can identify the Ki-67 antigen, which is present in the nucleus of cells during the G1, S, and G2 phases of the mitosis as well as cell division cycle [13,14,15]. Furthermore, we recently developed a novel scoring sheet using FibrinOgen, C-Reactive protein, Gender, and peritumoral Edema (FORGE score) as a tool to preoperatively estimate the MIB-1 index and the probability of PFS in cranial meningiomas [16,17]. Moreover, it was found that pro-tumor M2 macrophages are more common in both recurrent and WHO grade 2 cranial meningiomas [18]. However, the emerging role of inflammatory burden and macrophage infiltrates is unclear in the subgroup of spinal meningiomas so far. Anatomical location and the differentiation between skull base and non-skull base meningiomas were also found to be of paramount importance regarding tumor progression and proliferative potential [19,20]. Furthermore, patients with meningiomas present with various symptoms, which result from the compression of adjacent nervous structures and are dependent from the anatomic location. For instance, it has been revealed that patients with convexity meningiomas are more likely to present with seizures as baseline symptom compared to skull base meningiomas, whereas skull base meningiomas result in a higher frequency of headaches, anosmia, ocular deficits, and auditory deficits [21]. Moreover, degree and type of meningioma symptomatology have a significant influence on the quality of life [22].

Against this backdrop, we investigated our institutional series of Simpson grade I or II treated WHO grade 1 and 2 cranial and spinal meningiomas regarding demographic, clinical symptoms, proliferative potential, and macrophage infiltrates. Furthermore, those characteristics were investigated with regard to the probability of progression-free survival and the association with location-specific clinical deficits.

## 2. Materials and Methods

### 2.1. Study Design

Between January 2009 and July 2019, 880 patients underwent surgery for cranial WHO grade 1 and 2 meningioma in the institutional neurosurgical center. One-hundred-thirty patients were surgically treated for spinal WHO grade 1 and 2 meningioma between January 2000 and July 2020. A retrospective analysis of patient data was performed after institutional review board approval had been obtained.

### 2.2. Patient Selection

The inclusion criteria of this study were histopathologically confirmed meningioma, an age greater than 18 years, the availability of neuropathology reports, including MIB-1 index and mitotic count, preoperative systemic inflammatory parameters (fibrinogen or C-reactive protein), and treatment via a Simpson grade I or II resection. Patients who underwent a Simpson grade ≥III resection were excluded because partially or subtotally resected tumors do not necessarily contain the “hotspot area”, which displays the part of the tumor with the highest proliferative activity [23]. Furthermore, a homogeneous study cohort regarding extent of resection enables a sufficient analysis of progression-free survival. Neurofibromatosis type 2 associated meningiomas were excluded due to their differences regarding pathology and growth potential [24].

### 2.3. Data Recording

Clinical data including age, sex, neurological functioning, presence of symptomatic epilepsy, Modified McCormick (MMS) scale in spinal meningiomas, comorbidities, Karnofsky Performance Status (KPS), body mass index (BMI), anatomic location, WHO grading based on postoperative neuropathological investigation, immunohistochemical analysis, extent of meningioma resection based on Simpson grading system according to the European Association of Neuro-Oncology (EANO) (Simpson grade I–III = gross total resection, Simpson grade IV = subtotal resection, and Simpson grade V = biopsy), and postoperative follow-up data were saved into a computerized database (SPSS, version 27 for Windows, IBM Crop., Armonk, NY, USA) [16,25]. Modified McCormick scale was dichotomized into “good” (I + II) and “poor” (III–V) as described previously [26]. Anatomical location was dichotomized into cranial and spinal meningioma. Cranial meningiomas were further categorized into: (1) medial skull base tumors: olfactory groove, planum ethmoidale–sphenoidale, parasellar, tuberculum sellae, clival–petroclival, and foramen magnum meningiomas; (2) lateral skull base: lateral and middle sphenoid wing meningiomas, temporal fossa, spheno–orbital meningiomas, meningiomas of the petrous bone and occipital fossa; (3) non-skull base tumors: convexity, parasagittal, falx, tentorium, cerebellar convexity, pineal region, and intraventricular meningiomas [27]. A laboratory values collection was performed using the laboratory information software Lauris (version 17.06.21, Swisslab GmbH, Berlin, Germany). Collection of venous blood samples was routinely performed at a constant timepoint within 24 h prior to the surgical resection of cranial or spinal meningiomas. This standardized and constant time point makes it feasible to analyze the probabilities of progression-free survival. The routine laboratory protocol includes the following measures: complete blood count, kidney tests, liver tests, and coagulation profile (INR, aPTT). The baseline values of plasma fibrinogen were calculated by the Clauss method, which includes adding a standard and high concentration of thrombin (Dade^®^ thrombin reagent, Siemens Healthineers, Erlangen, Bavaria, Germany) to platelet poor plasma. Reference curves were used to determine the fibrinogen concentrations. The serum C-reactive protein values were obtained by turbidimetric immunoassays with a CRPL3 reagent (Roche, Basel, Switzerland) [16,17,28]. Surgical adverse events were also recorded. Surgical site infections (SSI) were identified using postoperative clinical notes based on the centers for disease control and prevention definition of infection (infection involving superficial incisional, deep incisional, and/or organ/space SSI). Definition of SSI was not limited to a period of 30 days after surgery. Hence, any SSI diagnosed at any time in the follow-up period were included. Furthermore, wound dehiscences without evidence of an SSI were not considered [29]. Furthermore, postoperative CSF fistulas and postoperative (intracerebral, subdural, and epidural) hemorrhage requiring revision surgery were also recorded.

### 2.4. Histopathology

Neuropathological grading was performed according to the 2016 WHO criteria [3]. All pathology reports were reviewed again to reconfirm that diagnosis was in line with these requirements. Immunohistochemistry was performed as described previously for paraffin-embedded biopsy tissue specimens [30,31]. The MIB-1 labeling index was determined using the following antibody: Anti-KI67 (Clone 2B11+PD7/26, DAKO, Glostrup, Denmark). Visualization was performed with diaminobenzidine, and neuropathological assessment was carried out by an expert neuropathologist (including AJB). The MIB-1 index was examined in randomly selected high-power microscopic fields. The proportions of stained and unstained nuclei in the neoplastic cells were determined. Furthermore, semiquantitative investigation and scoring of CD68^−^ stainings using anti-CD68 antibodies to detect macrophages was performed (Clone KP1, dilution 1:1000, DAKO, Glostrup, Denmark) [16,17]. Semi-quantitative scoring was performed by a senior neuropathologist and tumor specimen were investigated for the absence, focal, or diffuse staining of CD68 macrophages. The further histopathological workflows were as previously described [32].

### 2.5. Follow-Up

Imaging and clinical follow-up workflow includes the first postoperative MRI scans at 3 months after surgical treatment as well as on an annual basis for the ongoing years. Earlier clinical and imaging examinations were scheduled in case of new or progredient neurological deficits as well as radiological signs of meningioma progression [16,17]. Recurring meningiomas, which regrow at the local cavity of the primary surgery, were considered for analysis. The time to meningioma progression was defined as the time interval between the initial tumor resection and the initiation of a subsequent treatment based on radiological progression (e.g., secondary surgery or radiotherapy). Any radiological tumor progression without clinical importance, thus not resulting in any type of adjuvant therapy, was not considered. Furthermore, tumors growing at a location other than the primary meningioma site were not included for analysis [33].

### 2.6. Statistical Analysis

Data were organized and analyzed using SPSS for Windows (v27.0; IBM Crop, Armonk, NY, USA). Medial skull base, lateral skull base, non-skull base, and spinal meningiomas were analyzed regarding demographics, laboratory data, histopathology, location-specific neurologic deficits, and immunohistopathology using Fisher’s exact test (two-sided) and an analysis of variance (ANOVA). Cranial and spinal meningiomas were compared using Pearson’s chi-squared test (two-sided) and independent *t*-test after normal distribution was determined. Multivariable binary logistic regression analysis was performed to identify variables predicting increased MIB-1 index (≥5%) and density of CD68^+^ macrophage infiltrates. Dichotomizations of serum CRP (≤1.37 mg/L/>1.37 mg/L) and plasma fibrinogen (≤2.85 g/L/>2.85 g/L) in the prediction of MIB-1 were performed according to previously identified cut-off values for cranial meningiomas [16,17]. Age was dichotomized using the median-split [34]. Visualization of the results was supported by Prism 8 for macOS (Version 8.4.3, GraphPad Software, San Diego, CA, USA). Kaplan-Meier charts and log-rank tests of PFS were calculated. Uni- and multivariable Cox regression analyses were performed to investigate the PFS.

## 3. Results

### 3.1. Patient Characteristics

Five-hundred-and-forty-one patients underwent Simpson Grade I or II resection for intracranial or spinal meningioma. The results of the selection process are summarized in Figure 1.

Four-hundred-and-eighteen (418/541; 77.3%) patients had a cranial meningioma, whereas 123 (123/541; 22.7%) had a spinal meningioma. The median age (range) in the entire study cohort was 61.0 (23–91) years. Patients with a spinal meningioma were significantly older and presented with a poorer baseline KPS compared to the cranial meningioma patients. Furthermore, they had higher serum CRP and plasma fibrinogen levels compared to the cranial ones at the preoperative baseline laboratory examination. Regarding histopathology, WHO grade 2 meningiomas are scarce in the group of spinal meningiomas and both MIB-1 labeling index and number of mitotic figures are decreased compared to cranial meningiomas. Moreover, reduced density of CD68^+^ macrophage infiltrates was observed in the cohort of spinal meningiomas. Patient characteristics and further stratification by anatomic location are summarized in Table 1 and Table 2.

### 3.2. MIB-1 Index in Different Anatomic Locations of Meningioma

Mean +/− MIB-1 index in cranial meningiomas was 5.4 +/− 3.0, and 4.3 +/− 2.4 in spinal meningiomas, respectively (*p* = 0.007). Figure 2 displays the MIB-1 indices stratified by four locations. Non-skull base meningiomas had a significantly higher mean +/− SD MIB-1 index at 6.2 +/− 3.5. Further comparisons of mean MIB-1 labeling indices among the four groups of anatomic locations are presented in Table 3.

The median (range) MIB-1 index was 5% (1–20) in the study cohort. Multivariable binary logistic regression analysis, including age (>60/≤60), sex (female/male), serum CRP (≤1.37 mg/L/>1.37 mg/L), plasma fibrinogen (≤2.85 g/L/>2.85 g/L), WHO grade (1/2), and meningioma location (cranial/spinal), was performed to identify variables predicting an increased MIB-1 index (≥5%). Cranial meningioma (OR: 2.42, 95% CI: 1.51–3.86, *p* < 0.001) and WHO grade 2 (OR: 2.54, 95% CI: 1.40–4.60, *p* = 0.002) were identified as independent predictors of an increased MIB-1 index compared to spinal and WHO grade 1 meningiomas. Figure 3 displays the results of the multivariable analysis.

### 3.3. Density of CD68^+^ Macrophage Infiltrates

CD68^+^ staining was observed in 88 (71.5%) of the spinal meningioma group, and in 288 (68.9%) of the cranial meningiomas. Diffuse CD68^+^ staining was observed in 44 (50.0%) of the spinal meningiomas, whereas in the cranial meningioma group 208 patients had diffuse CD68^+^ macrophage infiltrates, respectively (Pearson´s chi-squared test (two-sided): *p* = 0.001). We conducted a multivariable binary logistic regression analysis to determine independent variables to be associated with diffuse CD68^+^ macrophage infiltrates. Multivariable analysis was performed with consideration of age (>60/≤60), sex (female/male), WHO grade (1/2), and meningioma location (cranial/spinal). The multivariable analysis revealed the variables “WHO grade 2” (OR: 2.05, 95% CI: 1.13–3.70, *p* = 0.02), and “cranial meningioma” (OR: 2.95, 95% CI: 1.77–4.91, *p* < 0.001) to be independent predictors for an increased density of CD68^+^ macrophage infiltrates. Figure 4 summarizes the results of the multivariable analysis.

Furthermore, Figure 5 displays two representative cases of the differences regarding MIB-1 index and density of CD68^+^ macrophages in different anatomical meningioma locations. The first case is a right-sided frontal convexity meningioma (Figure 5A–C) with both an increased MIB-1 index (10%) and diffuse infiltrates of CD68^+^ macrophages. The second case (Figure 5D–F) is a thoracic meningioma with a low MIB-1 index (2%) and only focal staining of CD68^+^ macrophages.

### 3.4. Anatomic Meningioma Location in the Prediction of Progression-Free Survival

MR-imaging follow-up was available in 465 patients (465/541; 86.0%). The mean (range) imaging follow-up time was 31.5 (3–169) months. Simpson grade II resections were performed in 132 (132/218; 60.6%), 67 (67/123; 53.7%), and 76 (76/200; 38.0%) patients in the group of skull base, spinal, and non-skull base meningiomas, respectively (results of the Pearson´s chi-squared analysis (two-sided): skull base vs. spinal: *p* = 0.33; skull base vs. non-skull base: *p* = 0.001, spinal vs. non-skull base: *p* = 0.005). The mean PFS time in the entire population was 145.7 months (95% CI: 134.9–156.6). Mean PFS times in the spinal meningioma group (n = 103), medial skull base meningioma group (n = 108), lateral skull base meningioma group (n = 81), and non-skull base meningioma group (n = 173) were 162.2 (95% CI: 150.5–174.0) months, 125.3 (95% CI: 117.0–133.5) months, 116.7 (95% CI: 104.3–129.2) months, and 88.7 (95% CI: 80.4–97.1) months, respectively (log-rank test: *p* = 0.07; see Figure 6A). In general, cranial meningiomas had a mean time to tumor progression of 115.9 (95% CI: 107.5–124.3) months compared to 162.2 (95% CI: 150.5–174.0) months in the spinal meningioma group. Based on the Kaplan–Meier method using log-rank test, spinal meningioma was significantly associated with enhanced PFS compared to cranial meningioma (*p* = 0.024, see Figure 6B).

Univariable Cox regression analysis revealed a significant association between cranial meningioma and tumor progression (hazard ratio (HR): 4.56, 95% CI: 1.07–19.37, *p* = 0.04). Univariable analysis showed also significant associations for both Simpson grade II resection (HR: 2.68, 95% CI: 1.19–6.00, *p* = 0.02) and WHO grade 2 (HR: 4.95, 95% CI: 2.31–10.62, *p* = 0.001) with shortened time to meningioma progression.

Multivariable Cox regression analysis of PFS with consideration of age (>60/≤60), sex (female/male), KPS (<90/≥90), meningioma location (cranial/spinal), Simpson grade (II/I), and WHO grade (2/1) was performed. The analysis demonstrated cranial meningioma as an independent statistically significant predictor for shortened PFS (HR: 4.71, 95% CI: 1.04–21.3, *p* = 0.04). Male sex (HR: 2.44, 95% CI: 1.06–5.64, *p* = 0.04), poor baseline KPS <90 (HR: 2.37, 95% CI: 1.05–5.33, *p* = 0.04), Simpson grade II (HR: 2.89, 95% CI: 1.24–6.76, *p* = 0.01), and WHO grade 2 (HR: 4.08, 95% CI: 1.84–9.05, *p* = 0.001) were also significant predictors for meningioma progression in this group of Simpson grade I or II resected WHO grade 1 and 2 meningiomas. Table 4 summarizes the results of the uni- and multivariable Cox regression analysis of PFS.

### 3.5. Clinical Implications of MIB-1 Index in Different Anatomical Locations

Meningiomas in close proximity to the optic nerve and optic tract, non-skull base meningiomas (i.e., convexity, parasagittal, and falx), and spinal meningiomas were analyzed regarding the association between MIB-1 index and corresponding location-specific clinical dysfunctions at presentation.

Sixty-seven meningiomas in close proximity to the optic nerve and optic tract were identified. The following (in descending order of frequency) anatomical meningioma locations affected the optic nerve and were included in the analysis: sphenoid wing (*n* = 34), olfactory groove (*n* = 15), planum sphenoidale (*n* = 8), tuberculum sellae (*n* = 5), and spheno–orbital meningioma (*n* = 5). Twenty-four (24/37; 64.9%) of the patients with reduced vision at presentation had an MIB-1 index ≥5%, compared to 10 (10/30; 33.3%) patients with an MIB-1 index <5% (*p* = 0.01).

Non-skull base meningiomas were evaluated regarding the association between MIB-1 index and epilepsy at presentation. Fifty (50/149; 33.6%) non-skull base meningioma patients with an MIB-1 index ≥5% had symptomatic epilepsy at presentation, compared to 9 (9/51; 17.6%) of the group with an MIB-1 index <5% (*p* = 0.03). Furthermore, 23 (23/116; 19.8%) skull base meningioma patients with an MIB-1 index ≥5% presented with symptomatic epilepsy, whereas 11 (11/102; 11.8%) of the skull base meningiomas patients with an MIB-1 index <5% had symptomatic epilepsy at presentation (*p* = 0.08). Among all cranial meningioma patients with an MIB-1 index ≥5%, symptomatic epilepsy at presentation was observed in 73 cases (73/265; 27.5%), and 21 (21/153; 13.7%) cranial meningioma patients with an MIB-1 index <5% had symptomatic epilepsy at baseline, respectively (*p* = 0.002).

Spinal meningiomas were investigated concerning the correlation between MIB-1 index and neurological functioning at presentation using the modified McCormick scale. Thirty (30/79; 38.0%) spinal meningioma patients with an MIB-1 index ≥5% had a poor MMS (III–V) at presentation, compared to 10 (10/44; 22.7%) patients with an MIB-1 index <5% (*p* = 0.08). Figure 7 summarizes the results of the analysis regarding the value of the MIB-1 index regarding location-specific clinical dysfunctions.

### 3.6. Postoperative Course and Postoperative Complications

Surgical site infections were observed in 16 cases (3.8%) in the cranial meningioma group, whereas only one SSI was found in the spinal meningioma group. The rates of revision surgeries due to postoperative hemorrhage or postoperative fistula were also higher in cranial meningiomas compared to spinal meningiomas. Spinal meningiomas had a significantly lower baseline KPS at admission, but the KPS did not differ among cranial and spinal meningiomas at 3- and 12-month follow-ups after surgery for meningioma. Mean (+/− SD) length of stay for surgical therapy of a cranial meningioma was 11.0 days (+/− 8.1), whereas spinal meningioma patients stayed for 13.1 days (+/−12.7%). New onset symptomatic epilepsy within the first 3 months after surgery was observed in 24 patients (6.0%) of the cranial meningioma group. Mean (+/− SD) MIB-1 labeling index in patients with new postoperative symptomatic epilepsy was 6.5 (+/− 2.8), whereas mean (+/− SD) MIB-1 index in patients without symptomatic epilepsy in the first 3 months after surgery was 5.3 +/− 3.0 (*p* = 0.07). MIB-1 labeling indices were further investigated with regard to the postoperative ambulatory mobility in spinal meningioma patients at 3 months after surgery. The Modified McCormick scale was available in 80 patients at the 3-month follow-up examination. Eight spinal meningioma patients (8/33; 24.2%) with an MIB-1 index ≥5% had a poor MMS (III–V), and 4 (4/47; 8.5%) spinal meningioma patients with an MIB-1 index < 5% had a poor MMS (III–V), respectively (*p* = 0.05). Further details are summarized in the Table 5.

## 4. Discussion

Known predictors for meningioma progression are the male sex, a young age at histopathological confirmation, poor Karnofsky Performance Status at diagnosis, WHO grade 2 or 3, increased mitotic count, Simpson grade ≥ III, and involvement of cranial nerves [35]. Increased MIB-1 indices are inversely associated with the time to tumor progression and correlate with the WHO grade of meningiomas [12,36,37]. Spinal meningiomas are suggested to grow more slowly compared to cranial meningiomas [38]. The pathophysiological mechanism and clinical implications regarding recurrence or progression are still not clearly understood. The present investigation demonstrates that spinal meningiomas have a distinct growth potential and meningioma microenvironment.

Our results can be summarized into four main points. (1) MIB-1 indices reflecting the proliferative activity and inflammatory tumor microenvironment are significantly lower in spinal meningiomas compared to cranial meningiomas. Spinal meningioma location is an independent predictor of a low MIB-1 index. (2) CD68^−^ staining depicting tumor-associated macrophages are significantly reduced in spinal meningiomas compared to cranial meningiomas. Spinal meningioma location and WHO grade 1 are independent predictors of a reduced density of macrophage infiltrates. (3) Spinal location of meningioma is significantly and independently associated with enhanced PFS. (4) High MIB-1 index is associated with location-specific clinical dysfunction at presentation, i.e., higher rates of seizure in convexity, parasagittal, and falcine meningiomas, and higher rates of cranial nerve dysfunction/visual loss in meningiomas in close proximity to the optic nerve and optic tract.

In the present investigation, we compared the demographic, laboratory, neuropathological, and immunohistochemical characteristics of cranial and spinal meningiomas. Number of mitotic figures and MIB-1 indices were significantly lower in spinal meningiomas compared to cranial meningiomas. Multivariable binary logistic regression analysis identified the spinal location of meningioma and WHO grade 1 as independently and significantly associated with low MIB-1 labeling indices (<5%). Already Roser et al. [38] revealed in a retrospective single-center series of 26 spinal meningiomas that the proliferative activity of spinal meningiomas is significantly lower compared to intracranial meningiomas. However, they could not find a significant difference regarding time to tumor progression between intracranial and spinal meningiomas. Maiuri et al. [27] also identified that spinal meningiomas have lower MIB-1 indices compared to skull base or non-skull base meningiomas. However, this retrospective investigation included a spinal meningioma cohort of 28 patients and only dichotomized MIB-1 indices were available. Proliferative activity is suggested to be inversely correlated with progesterone receptor expression. It was found that intracranial meningiomas with higher progesterone receptor expression have significantly lower MIB-1 indices [39,40,41,42,43]. Only a few studies have exclusively investigated this interesting pathophysiological pathway for spinal meningiomas but could not find a significant correlation of progesterone receptor expression with the MIB-1 index [38,44,45]. However, a recent study by Maiuri et al. [46] found a statistically significant increased expression of progesterone receptors in medial skull base and spinal meningiomas in a dichotomized comparison with lateral skull base and non-skull base meningiomas. In a previous institutional series, we identified that plasma fibrinogen levels and serum C-reactive protein are inversely associated with MIB-1 index in cranial meningiomas [16]. Univariable analysis showed that serum CRP values and plasma fibrinogen levels are also increased in the spinal meningiomas compared to cranial meningiomas, which would have been in line with the findings of a low FORGE-score in cranial meningiomas reflecting a reduced proliferative activity and estimating a lower MIB-1 index. However, multivariable analysis showed that serum CRP and plasma fibrinogen are not independently associated with the MIB-1 labeling index in a cohort also investigating spinal meningiomas. The role of systemic inflammatory burden is an emerging avenue in the treatment of meningiomas. The simple association between quick-to-use biomarkers such as serum CRP and tumor growth in meningiomas might be explained by the capability of human meningioma cells to secrete interleukin-6, which has an autocrine inhibitory role in the regulation of neoplastic cell growth [47]. Therefore, interleukin-6 secretion is inversely correlated with the growth velocity of meningiomas and both serum CRP and plasma fibrinogen are linked to the promotor of the interleukin-6 gene [48].

The present investigation also identified spinal location as an independent predictor of enhanced PFS. Spinal meningiomas are known to regrow less frequently compared to cranial ones. A recent review of 19 series of spinal meningiomas, including different Simpson grades and WHO grades, showed that the rate of recurrence ranges from 0% to 18% [45]. However, there are also investigations which showed no significant differences regarding tumor progression between intracranial and spinal meningiomas despite significantly lower MIB-1 indices in spinal meningiomas [38]. There are various theories debating the beneficial prognostic value of meningioma location along the spinal canal. Salpietro et al. [49] analyzed a retrospective series of 15 spinal meningiomas and suggested that spinal tumors do not invade the pial surface compared to the cranial meningiomas.

Several studies suggest that the resection of the dural origin is not significantly associated with reduced recurrence rates of spinal meningiomas [50,51]. However, our series focused on the biological behavior of the tumor tissue, which made it necessary to include only gross totally resected tumors in order to have a homogeneous cohort. In subtotally or partially resected meningiomas, the investigated neuropathological specimen does not necessarily reflect the areas with the highest proliferative potential [23]. Nakamura et al. [52] along with our results show that Simpson grade I resected spinal and cranial meningiomas have a significantly enhanced probability of PFS. Generally, Simpson grade I resection is suggested as the benchmark treatment in terms of recurrence-free survival in cranial meningiomas [53]. Despite gross total resection is the primary aim, surgical resection should be tailored to the individual patient according to preservation of neurological function and prevention of surgical adverse events. However, a recent meta-analysis of spinal meningiomas revealed no statistically significant differences between Simpson grade I or II regarding tumor recurrence [54]. Hence, Simpson grade I resection in spinal meningiomas must be considered carefully in preoperative treatment planning because of its strong association with cerebrospinal fluid fistulas following surgery, especially in ventrally located spinal meningiomas, which require complex dural reconstructions using artificial dura or thoracolumbar fascia [54].

Overall, physicians might consider a tailored follow-up and adjuvant therapy regime by considering both extent of resection as well as immunohistochemical data, such as MIB-1 labeling indices and CD68 staining for macrophages. A retrospective series comprising 239 surgically treated WHO grade 1 meningiomas showed a recurrence rate of 18.8% in patients who underwent a gross total resection and had MIB-1 labeling index > 4.5%. Those findings resulted in a similar risk for a meningioma recurrence as in patients who were treated by a subtotal resection. Hence, those results outline the paramount importance of a stringent follow-up of meningioma patients with an increased MIB-1 labeling index despite maximum cytoreductive surgery [55]. Moreover, adjuvant radiation treatment might be a potential therapy strategy in those patients with an increased MIB-1 labeling index or after partial resection of meningiomas. Subtotal resection followed by adjuvant radiation treatment of WHO grade 1 meningiomas has been shown to reduce the risk of recurrence [56,57,58,59]. However, most institutions still prefer follow-up imaging after subtotal resection [60]. Future scheduling of follow-up intervals might also benefit from a novel comprehensive approach by considering the MIB-1 index. The consideration of the MIB-1 index in the scheduling of follow-up intervals might be justified by the results of a prospective trial, which revealed that the MIB-1 labeling index is a reliable marker for the time to recurrence in WHO grade 1–3 meningiomas. This trial showed that patients with a MIB-1 labeling index between 0% and 4%, 5% and 9%, and ≥10% had 2.4, 4.9, and 9.7 recurrences per 100 person-years, respectively. Additionally, a MIB-1 labeling index ≥5% is a strong predictor for an early recurrence within the first two years after surgical treatment [61]. However, future and ongoing trials also have a special focus on the treatment effects of meningiomas and health-related quality of life is becoming more important. A recent population-based cohort study performed a survey with 104 spinal meningioma patients and had a response rate of 80.8%. The study revealed that before surgery, 48.8% were unable to work, and after surgery, all patients returned to work. Hence, this study strongly argues to consider surgical treatment in all patients with a spinal meningioma [62]. The results of this mentioned survey are in line with our results, which showed that the KPS and the ambulatory mobility of spinal meningioma patients increased after surgery. However, we also observed that spinal meningioma patients had a significantly lower baseline KPS compared to cranial meningioma patients. Spinal meningioma patients might have a lower baseline KPS because they are more often limited in their ambulatory ability due to compression of the spinal cord and the mean age at diagnosis was also higher in the spinal meningioma cohort. The lower baseline KPS and higher age at diagnosis of spinal meningioma patients might also explain the slightly longer length of stay compared to cranial meningioma patients. In addition, those patients often require a subsequent inpatient rehabilitation treatment.

In the present investigation, we found that spinal meningiomas have a significantly lower density of CD68^+^ macrophage infiltrates. Multivariable binary logistic regression analysis demonstrated that only spinal location and WHO grade 1 are independently associated with a reduced density of CD68^+^ macrophage infiltrates. In a recent immunohistochemical investigation of 30 cranial meningiomas by Proctor et al. [18], it was demonstrated that M2-macrophages have a pro-tumoral function and are predominantly present among the infiltrating tumor-associated macrophages. Moreover, it was identified that WHO grade 2 and recurrent meningiomas have significantly more M2-macrophages compared to primary meningiomas and WHO grade 1 tumors. Hence, pro-tumoral M2 macrophages are of paramount importance regarding the growth and recurrence of meningiomas. M1 macrophages might act tumoricidal by inducing inflammation via cytotoxic cytokine secretion and the recruitment of leukocytes, which are immunostimulant and impede the tumor growth [63,64,65]. Macrophage infiltrates account for 18% of all cells in meningiomas and their amount increases as the neuropathological grade of the tumor increases [66,67]. The composition of immune cells infiltrating the meningioma microenvironment might be of paramount importance regarding progression of the tumor and clinical functioning, such as seizure outcome and cranial nerve function. The present investigation and a previous institutional series showed that the MIB-1 index is significantly associated with both baseline cranial nerve function in skull base meningiomas and the new onset of cranial nerve deficits after surgery for frontobasal meningioma [32]. Furthermore, the burden of symptomatic epilepsy might also be a relevant clinical endpoint in patients with an increased MIB-1 index. Increased MIB-1 index was found to be significantly associated with symptomatic epilepsy at presentation in patients having convexity, parasagittal, or falcine meningiomas. MIB-1 index is an almost unknown variable among the determinants of symptomatic epilepsy in meningioma patients. Only one recent retrospective series investigating 384 meningioma patients revealed that an elevated MIB-1 index is an independent predictor of postsurgical seizure persistence [68]. Seizure is known to have a negative impact on health-related quality of life in meningioma patients [69]. Up to 25% of all meningioma patients present with seizure as the initial symptom [70]. Hence, the association between MIB-1 labeling index and seizure is of paramount importance to approach a novel target to improve both seizure burden and health-related quality of life in meningioma patients. Concerning the association between MIB-1 index and baseline neurological function in spinal meningiomas, we observed a trend towards a superior neurological functioning in patients with a low MIB-1 index. However, statistical results did not achieve a significance if threshold is set at <0.05. MIB-1 index significantly correlates with the density of CD68^+^ macrophages and the areas representing hotspots of increased MIB-1 indices are significantly overlapped by diffuse infiltrates of CD68^+^ macrophages [32]. Most of the macrophages within the infiltrates are suggested to be polarized to the M2 phenotype. However, meningiomas having a chromosome 22q deletion were found to predominantly have M1 phenotype macrophages [71]. Besides the importance of the polarization of the macrophages, Han et al. [72] demonstrated that meningioma patients with programmed death ligand-1 (PD-L1) expressing macrophages had a poorer survival rate. Therefore, targeting PD-L1 expressing macrophages might be a potential pathway in future immune therapy trials. PD-L1 expressing tumor cells might act in an inhibitory way on the activation of T-cells by binding to the PD-1 surface receptor of both T- and B-Cells [73]. Against this backdrop, there are several ongoing prospective trials analyzing the anti-PD1 antibodies avelumab, nivolumab, or pembrolizumab as new treatment options for meningiomas [74]. However, it must be remembered that CD68 staining does not enable a differentiation of the macrophage phenotypes. Furthermore, increased expression of CD68 was also selectively observed in tumors of rare subtype of meningiomas, such as the xanthomatous or histiocytic meningioma [75,76,77]. However, our study does not include one of those mentioned subtypes of meningiomas and the results show that spinal meningioma have a significantly reduced number of tumor-associated macrophages, which might imply that those meningiomas are not the primary collective for ongoing and future immunotherapeutic trials.

## 5. Conclusions

Spinal meningiomas show significantly lower MIB-1 indices reflecting a decreased growth rate. Meningioma location along the spinal location is significantly and independently associated with a reduced density of tumor-associated macrophages compared to intracranial meningiomas. Moreover, anatomic location along the spinal location is an independent prognostic factor for enhanced PFS. Ongoing and future trials targeting the inflammatory burden and microenvironment of meningiomas might focus on cranial meningiomas in order to slow tumor progression and improve the outcome of clinical endpoints such as cranial nerve function and seizure burden.

## Figures and Tables

**Figure 1 cancers-14-01033-f001:**
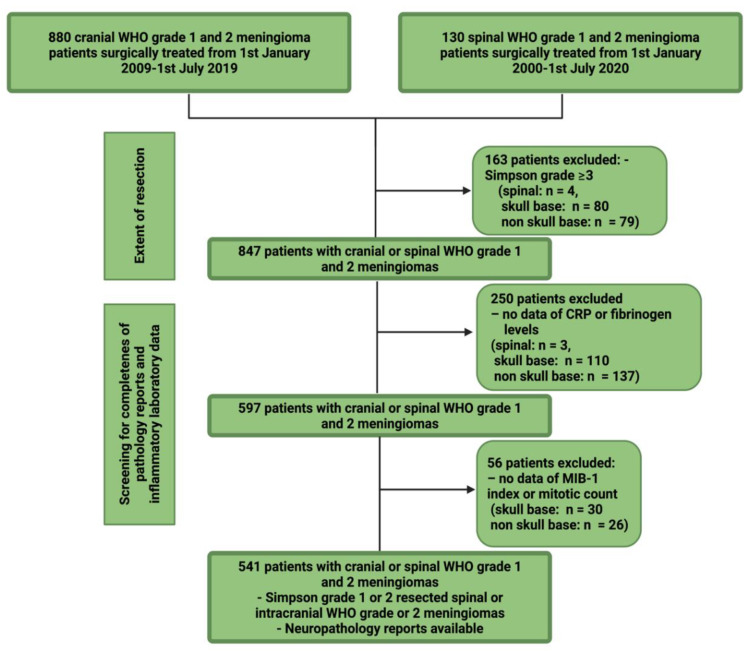
Flow chart illustrating the selection process of cranial und spinal meningioma patients in a single-center series.

**Figure 2 cancers-14-01033-f002:**
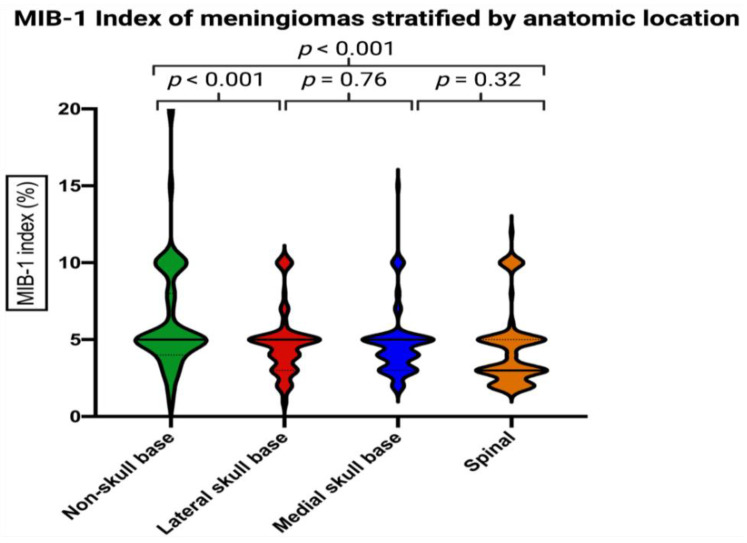
Violin plots displaying the MIB-1 indices in descending order among the anatomic locations. Violin plots shows mean and distribution of MIB-1 index. The thick horizontal black lines are the median values. *p*-values of the Student´s *t*-test are reported.

**Figure 3 cancers-14-01033-f003:**
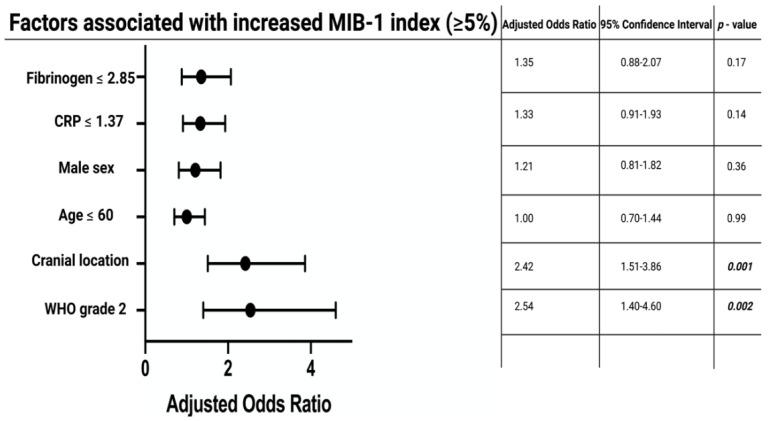
Forest plots from multivariable binary logistic regression analysis: cranial meningioma location, and WHO grade 2 are independent predictors of increased MIB-1 indices. *p*-values in italics and bold outline statistically significant results.

**Figure 4 cancers-14-01033-f004:**
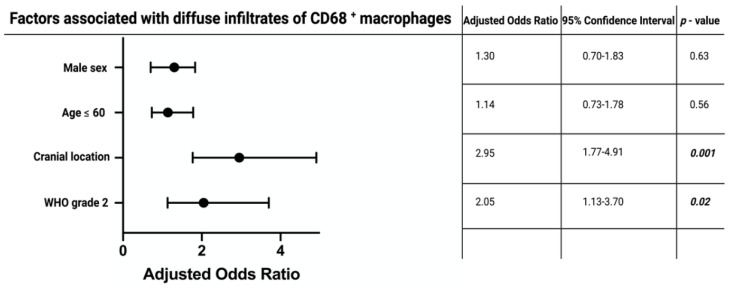
Forest plots from multivariable binary logistic regression analysis: cranial meningioma location, and WHO grade 2 are independent predictors of increased density of CD68^+^ macrophage infiltrates. *p*-values in italics and bold outline statistically significant results.

**Figure 5 cancers-14-01033-f005:**
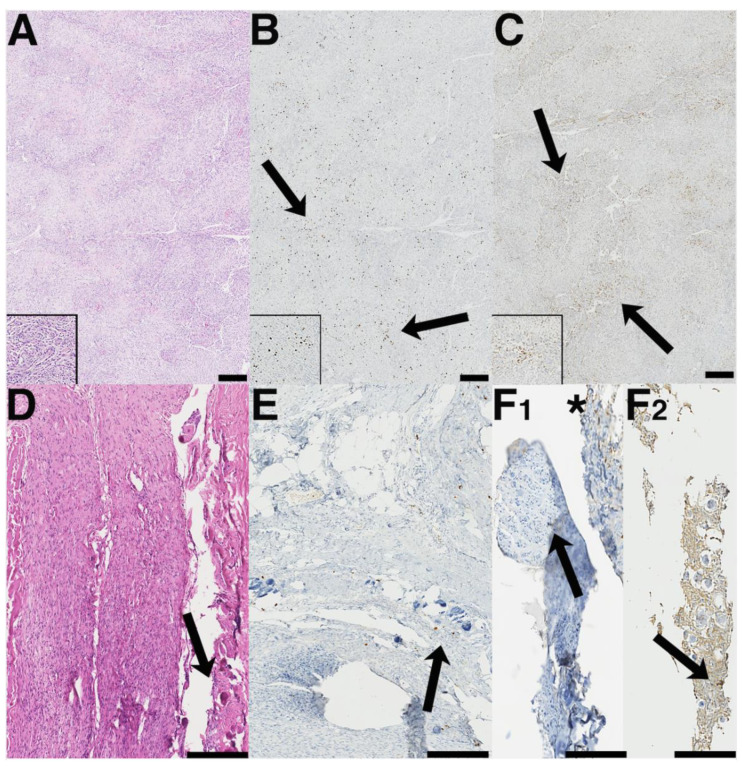
Representative images of two benign (meningothelial) meningiomas with fundamentally different Ki-67-labeling indices and macrophage infiltrates (**A**–**F**). (**A**) On the hematoxylin and eosin (HE) staining, meningothelial meningioma with high differentiation becomes apparent. Insert: note the whirl-like meningiomal structures that seem to become particular in the higher power magnification; cellularity is somewhat increased by non-meningioma, i.e., macrophagocytic cells. (**B**) Ki67-positive nuclei are substantially augmented in this tumor, both diffusely and in sometimes clustered distribution (black arrows). Insert demonstrates Ki67-positive nuclear clusters in the tumor. (**C**) The distribution of CD68^−^ positive macrophage infiltrates overlaps strongly with the areas of increased Ki67-labeling. The pattern of macrophage localization is overall diffuse-like with some interspersed clusters (black arrows). Insert: clustered macrophages in this meningioma. (**D**) Another meningioma with lobar growth pattern, syncytium-like appearance, and occasional Psammoma bodies (HE; black arrow). (**E**) Only a few Ki67-positive nuclei are present in this tumor, some of them accentuated in the periphery of Psammoma bodies (black arrow). CD68^−^ positive macrophage infiltrates are largely absent in solid meningioma compartments (black arrow in **F1**), as well as sparsely but not diffusely present in some areas (asterisk in **F1**); only occasional macrophages occur in the vicinity of Psammoma bodies (note the unspecifically higher background staining in **F2** in areas harboring Psammoma bodies; bar graphs in **A**–**E** are according to 100 μm and 50µm in **F1** and **F2** (inserts in **A**–**C** magnification as in **F1**).

**Figure 6 cancers-14-01033-f006:**
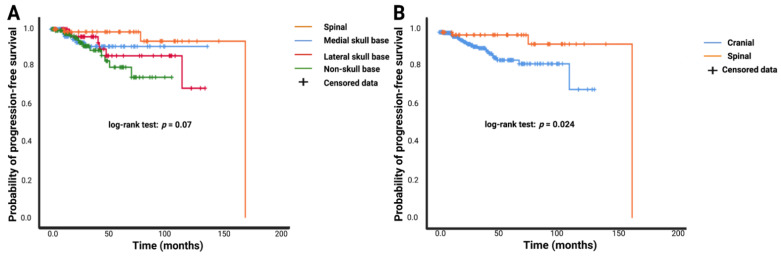
Kaplan–Meier curves of progression-free survival stratified by anatomic meningioma location. (**A**) Comparison of spinal (orange line), medial skull base (blue line), lateral skull base (red line), and non-skull base (green line). (**B**) Kaplan–Meier analysis of probability of time to meningioma progression stratified by cranial (blue line) versus spinal (orange line) meningioma. Censored data (= progression-free at last clinical examination) are represented by vertical dashes within all progression-free survival curves. The time axis is right-censored at 200 months. *p*-values of the log-rank test are reported.

**Figure 7 cancers-14-01033-f007:**
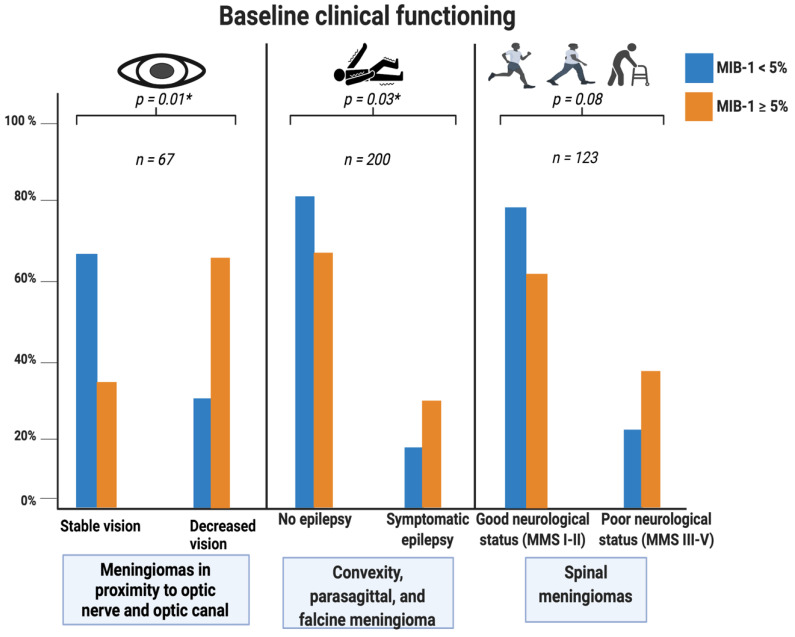
Grouped bar charts with percentages. Relative frequency of specific clinical dysfunctions at presentation stratified by anatomical location. Portion of meningioma patients having an MIB-1 index <5% is displayed by the blue bars, whereas the orange bars represent meningioma patients with an MIB-1 index ≥5%. *p*-values of the Pearson’s chi-squared test (two-sided) are reported. Significant *p*-values are marked with an asterisk.

**Table 1 cancers-14-01033-t001:** Patient characteristics among the groups of WHO grade 1 and 2 meningioma (*n* = 541). Statistical analysis was performed using independent *t*-test and Pearson’s chi-squared test (two-sided).

Variable	Cranial (*n* = 418)	Spinal (*n* = 123)	*p*-Value
Sex			0.09
Female	290 (69.4%)	95 (77.2%)	
Male	128 (30.6%)	28 (22.8%)	
Age			*0.001*
Mean +/− SD	60.7 +/− 13.6	65.6 +/− 12.4	
Baseline KPS			*0.001*
Mean +/− SD	89.8 +/− 11.5	82.1 +/− 11.0	
BMI			0.12
Mean +/− SD	27.3 +/− 5.9	28.3 +/− 5.1	
Serum CRP			0.04
Mean +/− SD	3.5 +/− 8.0	5.5 +/−9.8	
Plasma fibrinogen			0.003
Mean +/− SD	3.0 +/− 0.8	3.4 +/− 0.8	
WHO grade			*0.005*
1	344 (82.3%)	114 (92.7%)	
2	74 (17.7%)	9 (7.3%)	
Mitotic count			*0.002*
Mean +/− SD	1.9 +/− 2.8	1.1 +/− 1.2	
MIB-1 index			*0.007*
Mean +/− SD	5.4 +/− 3.0	4.3 +/− 2.4	
CD68^+^ staining (*n* = 376)			*0.001*
Focal	80 (27.8%)	44 (50.0%)	
Diffuse	208 (72.2%)	44 (50.0%)	
CD68^−^ staining (*n* = 165)	130 (31.1%)	35 (28.5%)	0.27

**Table 2 cancers-14-01033-t002:** Patient characteristics stratified by four major anatomic locations. Statistical analysis was performed using ANOVA and Fisher´s exact test (two-sided).

Variable	Non-Skull Base (*n* = 200)	Lateral Skull Base (*n* = 93)	Medial Skull Base (*n* = 125)	Spinal (*n* = 123)	*p*-Value
Sex					0.07
Female	136 (68.0%)	60 (64.5%)	95 (76.0%)	95 (77.2%)	
Male	64 (32.0%)	33 (35.5%)	30 (24.0%)	28 (22.8%)	
Age					*0.001*
Mean +/− SD	62.1 +/− 13.6	59.6 +/− 13.2	59.5 +/− 13.7	65.6 +/− 12.4	
Baseline KPS					*0.001*
Mean +/− SD	89.9 +/− 11.4	91.3 +/− 9.8	88.4 +/− 12.5	82.1 +/− 11.0	
BMI					0.18
Mean +/− SD	27.6 +/− 6.1	26.2 +/− 4.8	27.4 +/− 6.2	28.3 +/− 5.1	
Serum CRP					*0.03*
Mean +/− SD	4.3 +/− 10.9	2.5 +/− 3.1	2.9 +/− 3.8	5.5 +/− 9.8	
WHO grade					*0.007*
1	157 (78.5%)	79 (84.9%)	108 (86.4%)	114 (92.7%)	
2	43 (21.5%)	14 (15.1%)	17 (13.6%)	9 (7.3%)	
Mitotic count					*0.02*
Mean +/− SD	2.2 +/− 3.0	1.4 +/− 2.0	1.5 +/− 2.7	1.1 +/− 1.2	
MIB-1 index					*0.001*
Mean +/− SD	6.2 +/− 3.5	4.7 +/− 2.1	4.6 +/− 2.1	4.3 +/− 2.4	
CD68 staining (*n* = 376)					*0.001*
Focal	42 (25.0%)	18 (32.7%)	20 (27.5%)	44 (50.0%)	
Diffuse	111 (75.0%)	37 (67.3%)	60 (72.5%)	44 (50.0%)	

**Table 3 cancers-14-01033-t003:** Comparison of mean MIB-1 labeling indices among anatomic locations of meningioma.

Variable	Mean +/− SD	Mean Difference	95% CI of the Difference	*p*-Value
Non-skull base	6.2 +/− 3.5	1.48	0.83–2.13	*0.001*
vs.				
lateral skull base	4.7 +/− 2.1			
Non-skull base	6.2 +/− 3.5	1.57	0.96–2.18	*0.001*
vs.				
medial skull base	4.6 +/− 2.1			
Non-skull base	6.2 +/− 3.5	1.85	1.20–2.50	*0.001*
vs.				
spinal	4.3 +/− 2.4			
lateral skull base	4.7 +/− 2.1	0.09	−0.65–0.47	0.76
vs.				
medial skull base	4.6 +/− 2.1			
lateral skull base	4.7 +/− 2.1	0.37	−0.25–0.99	0.24
vs.				
spinal	4.3 +/− 2.4			
medial skull base	4.6 +/− 2.1	0.29	−0.28–0.85	0.32
vs.				
spinal	4.3 +/− 2.4			

**Table 4 cancers-14-01033-t004:** Uni- and multivariable Cox regression analysis of progression-free survival in WHO grade 1 and 2 meningiomas.

Variable	Univariable			Multivariable		
	HR	95% CI	*p*-Value	HR	95% CI	*p*-Value
Age	1.15	0.54–2.49	0.70	1.05	0.48–2.32	0.90
(**>60** vs. ≤60)						
Sex (**male** vs. female)	2.00	0.91–4.39	0.08	2.44	1.06–5.64	*0.04*
Karnofsky Performance Status (**<90** vs. ≥90)	2.00	0.94–4.26	0.07	2.37	1.05–5.33	*0.04*
Location (**Cranial** vs. spinal)	4.56	1.07–19.37	*0.04*	4.71	1.04–21.3	*0.04*
Simpson grade (**II** vs. I)	2.68	1.19–6.00	*0.02*	2.89	1.24–6.76	*0.01*
WHO grade (**2** vs. 1)	4.95	2.31–10.62	*0.001*	4.08	1.84–9.05	*0.001*

**Table 5 cancers-14-01033-t005:** Complications requiring revision surgery, length of stay and postoeprative course of physical status among cranial and spinal WHO grade 1 and 2 meningiomas (*n* = 541). Statistical analysis was performed using independent *t*-test and Pearson’s chi-squared test (two-sided).

Variable	Cranial (*n* = 418)	Spinal (*n* = 123)	*p*-Value
Surgical site infections	16 (3.8%)	1 (0.8%)	0.16
Postoperative hemorrhage	12 (2.8%)	2 (1.6%)	0.66
Postoperative CSF fistula	12 (2.8%)	3 (2.4%)	0.80
New onset epilepsy	24 (6.0%)	-	
Pulmonary embolism	5 (1.2%)	1 (0.8%)	0.72
Length of stay (in days)			
Mean +/− SD	11.0 +/− 8.1	13.1 +/− 12.7	0.08
Baseline KPS			*0.001*
Mean +/− SD	89.8 +/− 11.5	82.1 +/− 11.0	
KPS at 3 months			0.81
Mean +/− SD	88.7 +/− 17.9	88.1 +/− 10.0	
KPS at 12 months			0.54
Mean +/− SD	87.8 +/− 20.5	89.6 +/− 9.1	

## Data Availability

The data presented in this study are available on request form the corresponding author. The data are not publicly available due to privacy and ethical restrictions.

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
