# Peer review of "Proliferative Potential, and Inflammatory Tumor Microenvironment in Meningioma Correlate with Neurological Function at Presentation and Anatomical Location—From Convexity to Skull Base and Spine"

_cancers, 2022, doi:10.3390/cancers14041033_

Round 1
Reviewer 1 Report
Meningioma is the most frequent intracranial tumor and can be found in different locations in the skull and the spine. It has been shown that meningioma shows different histological features, symptoms and behavior depending on their location. The authors of this manuscript show again, using a retrospective analysis, that meningioma present different histological feature and behavior depending on their location, concentrating mostly in the difference between cranial and spinal meningioma. In my opinion, the manuscript contains, in its current state, several points that still need major improvement before a potential publication.
- Introduction:
- The introduction already contains a certain number of citations. However, a thorough review of the literature might be advisable regarding the impact of meningioma on patients’ health at presentation. Indeed, no mention of the frequency/type/importance… of neurological deficit at presentation is made while the authors use such parameters later in the article.
- Methods:
- A major concern is the decision by the authors to retrospectively compare two groups that were operated in different time-periods. Indeed, patients operated for cranial meningioma were included between 2009 and 2019 while patients operated for spinal meningioma were included between 2000 and 2020 (see text and Figure 1). Therefore, there was a twice bigger inclusion period for spinal meningioma compared to cranial meningioma. Such difference in inclusion periods could result in major bias and similar inclusion periods should be used between the two groups.
- Histopathology: a more detailed description of the quantitative and semi-quantitative analysis (of MIB-1, CD-68…) might be useful. Especially, as authors base a big quantity of their results on histopathological features.
- The authors decided to consider the “time to meningioma progression” as the interval until further treatment. However, in several articles, such timing is used for radiological considerations. Could the authors explain their choice further?
- A major point of discussion in this article is the inclusion of only Simpson 1 and 2 resection grade meningioma. Such inclusion is at risk to create an important selection bias as higher grade of Simpson resection are not so infrequent. Could the authors explain the rational for such inclusion criteria in the methodological part?
- Results:
- While the authors show several preoperative and histological features of the patients, there are (in general in this study), almost no mention of postoperative data such as postoperative complications or clinical evolution (except for the time to progression). Such information is however of critical importance as the maintenance (or improvement) of patients’ quality of life after meningioma surgery remains the main goal of surgery. Therefore, authors should include such information in this article. The same might be said for information such as duration of hospitalization.
- As the authors based their results on histopathological observations, a figure showing typical pictures of such observations (MIB and CD68 staining in the different locations) would be advisable to better define represent the authors’ results.
- Table 1: the authors show a proportion of skull-base VS non-skull base meningioma of approximately 50% VS 50%. Could they explain such results while the proportion in the literature often tends to be higher for non-skull base meningioma? Could these differences arise from the exclusion of a higher proportion of non-skull base meningioma patients? The characteristics of patients excluded (location…) should therefore also be mentioned in Figure 1 (or a table). Such observation can also be made for the number of spinal meningioma compared to cranial ones (as such a high proportion for spinal meningioma might originate from the inclusion criteria, therefore being less representative of a larger population).
- Figure 2: the statistical comparison between the different groups should be shown for all possible pairs (and not only for groups that are next to each other in this figure). Such results might be mentioned in another table.
- The authors compared the presence of epilepsy in non-skull base meningioma but not skull-base meningioma. Could they explain such decision?
- Figure 6: could the authors show the clinical evolution of patients’ symptoms after surgery? As mentioned earlier, such information is crucial during meningioma surgery.
- Discussion:
- A more in-depth discussion of the clinical significance of the authors’ finding (especially regarding eventual postoperative adjuvant therapy due to increased risk of progression depending on the histology) would be interesting. Could the presence of a different MIB/CD68 count advocate for another adjuvant therapy? Should patients’ follow-up be influenced by such histology (like it is the case in several institutions whether meningioma are graded as WHO 1 or 2)?
- As authors discuss the real benefit for gross total resection in spinal meningioma surgery, it would be advisable to also discuss the effect of meningioma treatment, with emphasis on the potential detrimental effects (neurological, psychological…) of such treatment on patients’ quality of life as this currently represents an important field of research.
Author Response
Dear Reviewer
Thank you for thoroughly reviewing our manuscript and the comments, which will allow us to improve it to a better scientific level and make it more understandable to the readership.
In the following we would like to respond to the remarks point-by-point:
“Introduction:
The introduction already contains a certain number of citations. However, a thorough review of the literature might be advisable regarding the impact of meningioma on patients’ health at presentation. Indeed, no mention of the frequency/type/importance… of neurological deficit at presentation is made while the authors use such parameters later in the article.“
Author´s response:
The reviewer is absolutely right that the association between meningioma location and symptoms at presentation is one of the essential objectives in this article. Therefore, we have revised the introduction according to this remark. Convexity meningioma patients are known to present significantly more often with symptomatic epilepsy as baseline symptom compared to skull base meningiomas, and skull base meningioma patients have more often anosmia, ocular deficits, and auditory deficits as baseline symptoms, respectively [1]. Furthermore, already the degree and kind of baseline symptomatology can have a significant impact on the long-term health-related quality of life in meningiomas [2].
References
- Wu, A.; Garcia, M.A.; Magill, S.T.; Chen, W.; Vasudevan, H.N.; Perry, A.; Theodosopoulos, P.V.; McDermott, M.W.; Braunstein, S.E.; Raleigh, D.R. Presenting Symptoms and Prognostic Factors for Symptomatic Outcomes Following Resection of Meningioma. World Neurosurg. 2018, 111, e149-e159
- Kim, S.R.; Shin, Y.S.; Kim, J.H.; Choi, M.; Yoo, S.H. Differences in Type Composition of Symptom Clusters as Predictors of Quality of Life in Patients with Meningioma and Glioma. World Neurosurg. 2017, 98, 50-59
“Methods: A major concern is the decision by the authors to retrospectively compare two groups that were operated in different time-periods. Indeed, patients operated for cranial meningioma were included between 2009 and 2019 while patients operated for spinal meningioma were included between 2000 and 2020 (see text and Figure 1). Therefore, there was a twice bigger inclusion period for spinal meningioma compared to cranial meningioma. Such difference in inclusion periods could result in major bias and similar inclusion periods should be used between the two groups.”
Histopathology: a more detailed description of the quantitative and semi-quantitative analysis (of MIB-1, CD-68…) might be useful. Especially, as authors base a big quantity of their results on histopathological features.
The authors decided to consider the “time to meningioma progression” as the interval until further treatment. However, in several articles, such timing is used for radiological considerations. Could the authors explain their choice further?
A major point of discussion in this article is the inclusion of only Simpson 1 and 2 resection grade meningioma. Such inclusion is at risk to create an important selection bias as higher grade of Simpson resection are not so infrequent. Could the authors explain the rational for such inclusion criteria in the methodological part?
Author´s response:
The reviewer is absolutely right that surgical techniques and inpatient treatment can vary among this long time period which was chosen for spinal meningiomas. However, the time period was chosen because spinal meningiomas represent a rare entity and account for only 12% of all meningiomas [1]. Hence, this “recruitment period” was necessary to enable a sufficient power size level for further statistical comparison between cranial and spinal meningiomas. Furthermore, the neuropathological grading was in line with the classification system of the 2021 WHO criteria [2]. The classification and grading of spinal meningiomas did not undergo substantial further revisions in 2016 and 2021 [3].
We agree with the reviewer regarding the description of histopathology and have revised the section 2.4. Semiquantitative investigation and scoring of CD68-stainings using anti-CD68 antibodies to detect macrophages was performed (Clone KP1, dilution 1:1000, DAKO, Glostrup, Denmark) [4, 5, 6]. Semi-quantitative scoring was performed by a senior neuropathologist (AJB) and tumor specimen were investigated for the absence, focal or diffuse staining of CD68+ macrophages. Furthermore, we provide a newly created figure 5 illustrating two typical cases. The first case shows the immunohistochemical staining using MIB-1 and CD68 in a right-sided frontal convexity meningioma with an increased MIB-1 labeling index and a diffuse infiltration of CD68+ macrophages. The second case represents a spinal meningioma located in the thoracic spine and shows a low MIB-1 labeling index with only focally expressed CD68+ macrophages.
The time to meningioma progression was defined as the time interval between the initial tumor resection and the initiation of any kind of subsequent therapy based on a radiological meningioma progression (e.g., radiotherapy or secondary surgery). Radiological tumor progressions without a clinical importance, which resulted in no type of adjuvant treatment were not considered. Moreover, only tumors regrowing at the local resection cavity of the primary surgery were considered for statistical analysis. Any tumor growing at another location and distant from the resection cavity of the primary meningioma was not included for analysis. This definition using a clinic-radiological approach was also used in a recent cohort study of 1469 consecutive meningioma patients, which analyzed the risk factors for recurrence and surgical adverse events of re-do surgery [7].
We included only patients who underwent a Simpson grade I or II resection because in partially or subtotally resected meningiomas, the tumor tissue does not necessarily contain the “hotspot area”, which represents the area with maximum proliferative activity [8]. Therefore, this inclusion criteria reduces the potential bias to underrate the proliferative and inflammatory potential of individual meningioma specimen. Furthermore, Simpson grade I or II resections as inclusion criteria enable a homogeneous cohort regarding the extent of resection and makes a reliable analysis of progression-free survival feasible. The reviewer is absolutely right that this information should be mentioned for the readership. Therefore, we added this explanation regarding those inclusion criteria in the methods.
References:
- Kwee, L.E.; Harhangi, B.S.; Ponne, G.A.; Kros, J.M.; Dirven, C.M.F.; Dammers, R. Spinal meningiomas: Treatment outcome and long-term follow-up. Clin Neurol Neurosurg. 2020, 198, 106238.
- Louis, D.N.; Perry, A.; Wesseling, P.; Brat, D.J.; Cree I.A.; Figarella-Branger, D.; Hawkins, C.; Ng, H.K.; Pfister, S.M.; Reifenberger, G.; Soffietti, R.; von Deimling, A.; Ellison, D.W. The 2021 WHO Classification of Tumors of the Central Nervous System: a summary. Neuro Oncol. 2021, 23(8), 1231-1251
- Louis, D.N.; Perry, A.; Reifenberger, G.; von Deimling, A.; Figarella-Branger, D.; Cavenee, W.K.; et al. The 2016 World Health Organization classification of tumors of the central nervous system: a summary. Acta Neuropathol.2016, 131, 803–20.
- Wach, J.; Lampmann, T.; Güresir, Á.; Schuss, P.; Vatter, H.; Herrlinger, U.; Becker, A.; Hölzel, M.; Toma, M.; Güresir, E. FORGE: A Novel Scoring System to Predict the MIB-1 Labeling Index in Intracranial Meningiomas. 2021, 13, 3643, doi:10.3390/cancers13143643.
- Wach, J.; Lampmann, T.; Güresir, Á.; Vatter, H.; Becker, A.; Hölzel, M.; Toma, M.; Güresir, E. Combining FORGE Score and Histopathological Diagnostic Criteria of Atypical Meningioma Enables Risk Stratification of Tumor Progression. 2021, 11(11), 2011, doi:10.3390/diagnostics11112011.
- Schneider, M.; Borger, V.; Güresir, A.; Becker, A.; Vatter, H.; Schuss, P.; Güresir, E. High Mib-1-score correlates with new cranial nerve deficits after surgery for frontal skull base meningioma. Neurosurg Rev. 2021, 44(1), 381-387.
- Lemée, J.M.; Corniola, M.V.; Meling, T.R. Benefits of re-do surgery for recurrent intracranial meningiomas. Sci Rep. 2020, 10(1), 303.
- Coons, S.W.; Johnson, P.C. Regional heterogeneity in the proliferative activity of human gliomas as measured by the Ki-67 labeling index. J Neuropathol Exp Neurol. 1993, 52, 609-618.
„Results:
- While the authors show several preoperative and histological features of the patients, there are (in general in this study), almost no mention of postoperative data such as postoperative complications or clinical evolution (except for the time to progression). Such information is however of critical importance as the maintenance (or improvement) of patients’ quality of life after meningioma surgery remains the main goal of surgery. Therefore, authors should include such information in this article. The same might be said for information such as duration of hospitalization.
- As the authors based their results on histopathological observations, a figure showing typical pictures of such observations (MIB and CD68 staining in the different locations) would be advisable to better define represent the authors’ results.
- Table 1: the authors show a proportion of skull-base VS non-skull base meningioma of approximately 50% VS 50%. Could they explain such results while the proportion in the literature often tends to be higher for non-skull base meningioma? Could these differences arise from the exclusion of a higher proportion of non-skull base meningioma patients? The characteristics of patients excluded (location…) should therefore also be mentioned in Figure 1 (or a table). Such observation can also be made for the number of spinal meningioma compared to cranial ones (as such a high proportion for spinal meningioma might originate from the inclusion criteria, therefore being less representative of a larger population).
- Figure 2: the statistical comparison between the different groups should be shown for all possible pairs (and not only for groups that are next to each other in this figure). Such results might be mentioned in another table.
- The authors compared the presence of epilepsy in non-skull base meningioma but not skull-base meningioma. Could they explain such decision?
- Figure 6: could the authors show the clinical evolution of patients’ symptoms after surgery? As mentioned earlier, such information is crucial during meningioma surgery.“
Author´s response:
We absolutely agree with the reviewer that postoperative outcome, length of stay and complications are essential regarding quality of life. Consequently, we added a newly created section “3.6 postoperative course and postoperative complications” to the results. Furthermore, a newly created table 5 is provided and summarizes the results. The table 5 includes data regarding the course of KPS until 12 months after surgery, surgical adverse events (e.g., surgical site infections, postoperative hemorrhage, postoperative CSF fistula), frequency of pulmonary embolism, new onset epilepsy after surgery, and length of stay (in days). The frequencies of surgical site infections, postoperative hemorrhage, and CSF fistulas were higher in the group of cranial meningiomas compared to the spinal meningioma group. However, the results did not achieve a statistical significance. The length of stay (mean +/- SD) among the spinal meningioma patients was 13.1 +/- 12.7 days, whereas cranial meningioma patients had a mean length of stay for 11.0 +/- 8.1 days (p = 0.008). This difference might be explained by the fact that the spinal meningioma patients were significantly older compared to cranial meningioma patients and they also had a lower baseline KPS which might be caused by a reduced ambulatory mobility due to the compression of the spinal cord necessitating surgical resection.
We also revised the section results regarding the analysis of immunohistochemical staining and provide a newly created figure 5 illustrating two typical cases. The first case (Fig. 5 A-C) shows a right-sided frontal convexity meningothelial meningioma with Ki67-positive nuclei being substantially augmented (black arrows) in this meningioma. Additionally, the distribution of CD68+ macrophage infiltrates significantly overlap with the areas of increased Ki-67 labeling in this convexity meningioma. Furthermore, the pattern of CD68+ macrophage infiltrates is diffuse with some interspersed clusters (black arrows) The second case (Fig. 5 D-F) represents a spinal meningioma located in the thoracic spine and shows a low MIB-1 labeling index with only focally expressed CD68+ macrophages (black arrows). The tumor appears syncytium-like and has a lobar growth pattern. There are only few Ki67-positive nuclei in this spinal meningioma, and they are accentuated in the periphery of psammoma bodies. The CD68+ are largely absent in the solid meningioma compartments and also sparsely but not diffusely present in some areas.
The reviewer is absolutely right that meningiomas predominantly grow at the convexity, parasagittal, or falx if the frequency is analyzed from an epidemiological point of view [1]. However, also other recent large study cohorts which investigated surgically treated intracranial meningiomas had a proportion of skull-base vs. non-skull base meningiomas of approximately 50:50 %. For instance, Meling et al. [2] analyzed surgically treated cranial meningiomas between 1990 and 2010. They identified 562 (49%) skull base meningiomas, and 586 (51%) non-skull base meningiomas, respectively. Meningiomas located on the convexity are usually accessible, circumscribed, and well demarcated from neural and vascular structures, whereas skull base meningiomas are often extensive, invasive, and intimately associated with cranial nerves, vessels and brainstem [3]. Skull base meningiomas often comprising a minority of completely asymptomatic meningiomas [4]. Moreover, it was found that the incidence of preoperative neurological deficits is often higher in skull base meningiomas [2]. Therefore, this might be the reason for the shift towards a higher proportion of skull base meningiomas in clinical series who analyzed solely surgically treated meningiomas. Nevertheless, we implemented the suggestion to show the anatomic location of meningiomas who were excluded during the selection process. Hence, the figure 1 was revised according to this remark.
The statistical analysis of the mean values of the MIB-1 index in the anatomical groups is now provided for all pairs of groups and is summarized in a newly created table 3 in the section “3.2 MIB-1 Index in different anatomic locations of meningioma”.
Convexity meningioma patients are known to present significantly more often with symptomatic epilepsy as baseline symptom compared to skull base meningiomas, and skull base meningioma patients have more often anosmia, ocular deficits, and auditory deficits as baseline symptoms, respectively [5]. However, the reviewer is absolutely right that also skull base meningiomas with a significant perilesional edema can cause a preoperative symptomatic epilepsy. Therefore, we have revised the section “3.5 Clinical implications of MIB-1 index in different anatomical locations” and provide the proportions of patients with a symptomatic epilepsy in patients with a MIB-1 index ≥ 5% and a MIB-1 index < 5% among skull base meningiomas, non-skull base meningiomas and all cranial meningiomas. MIB-1 ≥ 5% index was significantly associated with a symptomatic epilepsy at presentation in convexity, parasagittal, and falcine meningiomas. Twenty-three (23/116; 19.8%) skull base meningioma patients with a MIB-1 index ≥ 5% presented with a symptomatic epilepsy, whereas 11 (12/102; 11.8%) of the skull base meningiomas patients with a MIB-1 index < 5% had a symptomatic epilepsy at presentation (p = 0.08). Hence, this issue might be of special interest in non-skull base meningiomas. However, MIB-1 index > 5% was significantly associated with a symptomatic epilepsy in the entire cohort of all cranial meningiomas.
The reviewer is absolutely right that postoperative outcome can guide the intraoperative surgical decision making. In two previous investigations we approached this issue. We created a novel scoring system to preoperatively estimate the MIB-1 index in cranial meningiomas [8]. This information could be of paramount importance because intraoperative methods (e.g., rapid immunohistochemistry based on alternating current electric fields [6, 7]) to determine immunohistochemical characteristics are not established in the clinical workflow so far. Furthermore, we already showed that the MIB-1 labeling index ≥ 5% is associated with the postoperative onset of new cranial nerve deficits after surgery for frontal skull base meningiomas [9]. Hence, this score estimating the MIB-1 index and the knowledge that an increased MIB-1 index is independently associated with an increased prevalence of new cranial nerve deficits might outrank a maximum cytoreductive surgery for frontal skull base meningiomas which is also an independent risk factor for new cranial nerve deficits. Furthermore, we provide the frequency of new onset epilepsy within three months after surgery for cranial meningiomas. New onset symptomatic epilepsy within the first 3 months after surgery was observed in 24 patients (6.0%) of the cranial meningioma group. Mean (+/- SD) MIB-1 labeling index in patients with a new postoperative symptomatic epilepsy was 6.5 (+/- 2.8), whereas mean (+/- SD) MIB-1 index in patients without a symptomatic epilepsy in the first 3 months after surgery was 5.3 +/- 3.0 (p = 0.07). We strive to further investigate this issue regarding MIB-1 index and long-term epilepsy outcome in a further investigation. MIB-1 labeling indices was also further investigated with regard to the postoperative ambulatory mobility in spinal meningioma patients at 3-months after surgery. Modified McCormick scale was available in 80 patients at 3-months follow-up examination. Eight spinal meningioma patients (8/33; 24.2%) with a MIB-1 index ≥5% had a poor MMS (III-V), and 4 (4/47; 8.5%) spinal meningioma patients with a MIB-1 index <5% had a poor MMS (III-V), respectively (p = 0.05).
References
- Huntoon, K.; Toland, A.M.S.; Dahiya, S. Meningioma: A Review of Clinicopathological and Molecular Aspects. Front Oncol. 2020, 10, 579599.
- Meling, T.R.; Da Broi, M.; Scheie, D.; Helseth, E. Meningiomas: skull base versus non-skull base. Neurosurg Rev. 2019, 42(1), 163-173
- Nanda, A.; Vannemreddy, P. Recurrence and outcome in skull base meningiomas: do they differ from other intracranial meningiomas? Skull Base. 2008, 18(4), 243-52. doi: 10.1055/s-2007-1016956
- Islim, A.I.; Mohan, M.; Moon, R.D.C.; Srikandarajah, N.; Mills, S.J.; Brodbelt, A.R.; Jenkinson, M.D. Incidental intracranial meningiomas: a systematic review and meta-analysis of prognostic factors and outcomes. J Neurooncol. 2019, 142(2), 211-221.
- Wu, A.; Garcia, M.A.; Magill, S.T.; Chen, W.; Vasudevan, H.N.; Perry, A.; Theodosopoulos, P.V.; McDermott, M.W.; Braunstein, S.E.; Raleigh, D.R. Presenting Symptoms and Prognostic Factors for Symptomatic Outcomes Following Resection of Meningioma. World Neurosurg. 2018, 111, e149-e159
- Moriya, J., Tanino, M.A., Takenami, T., Endoh, T., Urushido, M., Kato, Y., Wang, L., Kimura, T., Tsuda, M., Nishihara, H., et al. Rapid immunocytochemistry based on alternating current electric field using squash smear preparation of central nervous system tumors.Brain Tumor Pathol. 2016, 33, 13–18.
- Terata, K., Saito, H., Nanjo, H., Hiroshima, Y., Ito, S., Narita, K., Akagami, Y., Nakamura, R., Konno, H., Ito, A., et al. Novel rapid-immunohistochemistry using an alternating current electric field for intraoperative diagnosis of sentinel lymph nodes in breast cancer. Rep. 2017, 7, 2810.
- Wach, J.; Lampmann, T.; Güresir, Á.; Schuss, P.; Vatter, H.; Herrlinger, U.; Becker, A.; Hölzel, M.; Toma, M.; Güresir, E. FORGE: A Novel Scoring System to Predict the MIB-1 Labeling Index in Intracranial Meningiomas. 2021, 13, 3643
- Schneider, M.; Borger, V.; Güresir, A.; Becker, A.; Vatter, H.; Schuss, P.; Güresir, E. High Mib-1-score correlates with new cranial nerve deficits after surgery for frontal skull base meningioma. Neurosurg Rev. 2021, 44(1), 381-387.
„Discussion:
- A more in-depth discussion of the clinical significance of the authors’ finding (especially regarding eventual postoperative adjuvant therapy due to increased risk of progression depending on the histology) would be interesting. Could the presence of a different MIB/CD68 count advocate for another adjuvant therapy? Should patients’ follow-up be influenced by such histology (like it is the case in several institutions whether meningioma are graded as WHO 1 or 2)?
- As authors discuss the real benefit for gross total resection in spinal meningioma surgery, it would be advisable to also discuss the effect of meningioma treatment, with emphasis on the potential detrimental effects (neurological, psychological…) of such treatment on patients’ quality of life as this currently represents an important field of research.“
Author´s response:
The reviewer is absolutely right regarding the remark about the discussion. Therefore, we have revised this section and extended the discussion about follow-up scheduling, adjuvant therapy regime, and influence on health-related quality of life. The MIB-1 index and the burden of macrophage infiltrates have relevant clinical implications regarding tailored scheduling of follow-up intervals. Hence, follow-up interval and adjuvant therapy regimes might be determined by considering extent of resection as well as immunohistochemical data such as MIB-1 labeling index and density of CD68+ macrophage infiltrates. For instance, a retrospective investigation of 239 WHO grade 1 meningioma patients who underwent surgery showed that patients with a gross total resection and a MIB-1 index >4.5% had a recurrence rate of 18.8%. Furthermore, patients who underwent a subtotal resection had a similar risk of recurrence in this mentioned study. Hence, those findings show the importance of a stringent and tailored follow-up schedule in patients with an increased MIB-1 labeling index despite maximum cytoreductive surgery was performed [1]. Furthermore, adjuvant radiation therapy might be a potential treatment avenue in those meningioma patients with an increased MIB-1 labeling index or after partial resection of meningiomas. Despite most institutions still perform follow-up imaging after subtotal resection [2], several investigations have shown that subtotal resection followed by adjuvant radiation treatment for WHO grade 1 meningioma reduces the rate of recurrence [3-6]. Additionally, it has been found that the MIB-1 labeling index is a sufficient marker for the time to recurrence in WHO grade 1-3 meningiomas and might enable the identification of an early recurrence within the first two years after surgery [7]. This prospective investigation showed revealed that patients with a MIB-1 labeling index of 0% to 4%, 5% to 9%, and ≥10% had 2.4, 4.9, and 9.7 recurrences per 100 person-years, respectively. The issue regarding surgical treatment and health-related quality of life is increasingly debated in the literature. A recent population-based cohort investigation performed a survey among surgically treated spinal meningiomas (n =104, response rate: 80.8%) and revealed that nearly half of all patients were unable to work before surgery, but all patients returned to work after surgical treatment [8]. Furthermore, the impact of the MIB-1 index on the presence of a symptomatic epilepsy might be essential for future investigations. Symptomatic epilepsy affects up to 25% of all meningioma patients at diagnosis [9]. Health-related quality of life is known to be negatively influenced by the presence of a symptomatic epilepsy in meningioma patients [10].
References
- Haddad, A.F.; Young, J.S.; Kanungo, I.; Sudhir, S.; Chen, J.S.; Raleigh, D.R.; Magill, S.T.; McDermott, M.W.; Aghi, M.K. WHO Grade I Meningioma Recurrence: Identifying High Risk Patients Using Histopathological Features and the MIB-1 Index. Front Oncol. 2020, 10, 1522.
- Rogers, L.; Barani, I; Chamberlain, M.; Kaley, T.J.; McDermott, M.; Raizer, J.; Schiff, D.; Weber, D.C.; Wen, P.Y.; Vogelbaum, M.A. Meningiomas: knowledge base, treatment outcomes, and uncertainties. A RANO review. J Neurosurg. 2015, 122(1), 4-23.
- Ohba, S.; Kobayashi, M.; Horiguchi, T.; Onozuka, S.; Yoshida, K.; Ohira, T.; Kawase, T. Long-term surgical outcome and biological prognostic factors in patients with skull base meningiomas. J Neurosurg. 2011, 114(5), 1278-87.
- Soyuer, S.; Chang, E.L.; Selek, U.; Shi, W.; Maor, M.H.; DeMonte, F. Radiotherapy after surgery for benign cerebral meningioma. Radiother Oncol. 2004, 71(1), 85-90.
- Park, S.; Cha, Y.J.; Suh, S.H.; Lee, I.J.; Lee, K.S.; Hong, C.K.; Kim, J.W. Risk group-adapted adjuvant radiotherapy for WHO grade I and II skull base meningioma. J Cancer Res Clin Oncol. 2019, 145(5), 1351-1360.
- Oya, S.; Ikawa, F.; Ichihara, N.; Wanibuchi, M.; Akiyama, Y.; Nakatomi, H.; Mikuni, N.; Narita, Y. Effect of adjuvant radiotherapy after subtotal resection for WHO grade I meningioma: a propensity score matching analysis of the Brain Tumor Registry of Japan. J Neurooncol. 2021, 153(2), 351-360.
- Mirian, C.; Skyrman, S.; Bartek, J. Jr.; Jensen, L.R.; Kihlström, L.; Förander, P.; Orrego, A.; Mathiesen, T. The Ki-67 Proliferation Index as a Marker of Time to Recurrence in Intracranial Meningioma. Neurosurgery. 2020, 87(6), 1289-1298.
- Pettersson-Segerlind, J.; von Vogelsang, A.C.; Fletcher-Sandersjöö, A.; Tatter, C.; Mathiesen, T.; Edström, E.; Elmi-Terander, A. Health-Related Quality of Life and Return to Work after Surgery for Spinal Meningioma: A Population-Based Cohort Study. Cancers (Basel). 2021, 13(24), 6371
- Lieu, A.S.; Howng, S.L. Intracranial meningiomas and epilepsy: incidence, prognosis and influencing factors. Epilepsy Res. 2000, 38(1), 45-52.
- Waagemans, M.L.; van Nieuwenhuizen, D.; Dijkstra, M.; Wumkes, M.; Dirven, C.M.; Leenstra, S.; Reijneveld, J.C.; Klein, M.; Stalpers, L.J. Long-term impact of cognitive deficits and epilepsy on quality of life in patients with low-grade meningiomas. Neurosurgery. 2011, 69(1), 72-8; discussion 78-9.
Reviewer 2 Report
Major points:
- Study design and patient characteristics: Please divide the paragraph into study design and patient selection and implement the patient characteristics in the results part, as this is a finding.
- Why did you include the mentioned variables in the multivariate analysis? What was you cut-off value for implementation?
- Is there a difference of extend of resection (Simpson grade) in different tumor localization? As dural resection is more challenging in skull base and spinal meningiomas, I expect to find more Simpson grade 2 in these cases. Therefore, this could also influence the PFS, although the recent meta-analysis did not show any differences in tumor recurrence.
- You concluded that MIB-1 index correlates with clinical symptoms. Can this also be due to different localization of the meningioma, e.g. optic nerve sheet meningioma is more likely to cause visual problems than sphenoid wing m.?
- The discussion part is hard to read and understand. Please revise this part and try to improve the structure.
Minor points:
- 3, l. 113-116: Please revise sentences, as context is hard to understand.
- 4, l. 148: “)” missing
- Table 1 has to be divided into two tables
- Figure 5. Please adapt the colors in the figure, e.g. spinal meningiomas in the same color in A and B.
Author Response
Dear Reviewer
Thank you for thoroughly reviewing our manuscript and the comments, which will allow us to improve it to a better scientific level and make it more understandable to the readership.
In the following we would like to respond to the remarks point-by-point:
Major points:
“Study design and patient characteristics: Please divide the paragraph into study design and patient selection and implement the patient characteristics in the results part, as this is a finding.”
Author´s response:
The reviewer is absolutely right that the figure 1 which displays the results of the selection process should be placed in the section “results”. Therefore, we have added the figure 1 to section results and revised the order in the methods part by dividing the section “2.1 study design and patient characteristics” into “2.1 Study design” and “2.2 Patient selection”.
“Why did you include the mentioned variables in the multivariate analysis? What was you cut-off value for implementation?”
Author´s response:
We agree with the reviewer that the cut-off values of the included variables have to be further explained. In a previous institutional investigation analyzing independent predictors of an increased MIB-1 labeling index in cranial WHO grade 1 and 2 meningiomas we have identified the following variables to be significantly associated with an elevated MIB-1 labeling index: Male sex, serum c-reactive protein levels, plasma fibrinogen levels, and peritumoral edema [1]. Based on those findings we have created a scoring system to preoperatively predict the MIB-1 labeling index in cranial WHO grade 1 and 2 meningiomas. Therefore, we have included those variables with the already identified optimum cut- off value for the present investigation. However, peritumoral brain edema was not included because in the present series we performed a multivariable analysis of predictors of MIB-1 labeling index for cranial and spinal meningiomas. Age was also included as a potential independent variable being associated with the MIB-1 labeling index because of the already described inverse correlation between MIB-1 labeling index and age. Matsuno et al. [2] investigated a series of 127 cranial meningiomas and revealed that patients below 40 years had a mean MIB-1 labeling index of 7.4%, whereas patients older than 40 years had a mean MIB-1 labeling index of 2.9%. The median age of all (cranial & spinal) meningioma patients in the present investigation was 61 years. Therefore, we applied a variable-oriented approach (i.e., median-split) which resulted in a dichotomization into patients ≤60 and >60 years [3]. Furthermore, WHO grade 2 was included because of the known strong correlation between MIB-1 labeling index with WHO grade of meningiomas. Babu et al. analyzed both cranial and spinal meningiomas regarding the association between MIB-1 labeling index and WHO grade in 300 cases. The mean MIB-1 index statistically significant increased from WHO grade 1 to 2 and from grade 2 to 3 [4]. Consequently, we have implemented those criteria in the section “2.6 Statistical analysis” in order to make it more understandable to the readership.
References:
- Wach, J.; Lampmann, T.; Güresir, Á.; Schuss, P.; Vatter, H.; Herrlinger, U.; Becker, A.; Hölzel, M.; Toma, M.; Güresir, E. FORGE: A Novel Scoring System to Predict the MIB-1 Labeling Index in Intracranial Meningiomas. 2021, 13, 3643, doi:10.3390/cancers13143643
- Matsuno, A.; Fujimaki, T.; Sasaki, T.; Nagashima, T.; Ide, T.; Asai, A.; Matsuura, R.; Utsunomiya, H.; Kirino, T. Clinical and histopathological analysis of proliferative potentials of recurrent and non-recurrent meningiomas. Acta Neuropathol. 1996, 91(5), 504-510
- Haller, B.; Ulm, K.; Hapfelmeier, A. A Simulation Study Comparing Different Statistical Approaches for the Identification of Predictive Biomarkers. Comput Math Methods Med. 2019, 2019, 7037230
- Babu, S.; Uppin, S.G.; Panigrahi, M.K.; Saradhi, V.; Bhattacharjee, S.; Sahu, B.P.; Purohit, A.K.; Challa, S. Meningiomas: correlation of Ki67 with histological grade. Neurol India. 2011, 59(2), 204-207
“Is there a difference of extend of resection (Simpson grade) in different tumor localization? As dural resection is more challenging in skull base and spinal meningiomas, I expect to find more Simpson grade 2 in these cases. Therefore, this could also influence the PFS, although the recent meta-analysis did not show any differences in tumor recurrence.”
Author´s response:
The reviewer is absolutely right that Simpson grade II resections were more frequent among the group of skull base or spinal meningiomas compared to non-skull base meningiomas. Therefore, we further added the rates of Simpson grade II resections to the section “3.4 Anatomic meningioma location in the prediction of progression-free survival”. The table 4 which summarizes both the univariable and multivariable analysis of PFS includes the extent of resection (Simpson grade I vs. II). Simpson grade II resection (Hazard Ratio: 2.89, 95% CI: 1.24-6.76, p = 0.01) was found as an independent predictor of meningioma progression in our series of cranial and spinal WHO grade 1 and 2 meningioma patients who underwent either Simpson grade I or II resections. Simpson grade II resections were performed in 132 (132/218; 60.6%), 67 (67/123; 53.7%), and 76 (76/200; 38.0%) patients in the group of skull base, spinal, and non-skull base meningiomas, respectively (Results of the Pearson´s chi-squared analysis (two-sided): skull base vs. spinal: p = 0.33; skull base vs. non-skull base: p = 0.001, spinal vs. non-skull base: p = 0.005).
“You concluded that MIB-1 index correlates with clinical symptoms. Can this also be due to different localization of the meningioma, e.g. optic nerve sheet meningioma is more likely to cause visual problems than sphenoid wing m.?”
Author´s response:
We absolutely agree with the reviewer that optic nerve sheath meningiomas will more likely and earlier cause decreased vision compared to conventional medial skull base meningiomas which slowly grow into the proximity to optic nerve and optic canal. In the present institutional series we identified no optic nerve sheath meningioma. The association between MIB-1 labeling index and visual function in optic nerve sheath meningioma would be of potential interest in a future series. However, this objective might have to be investigated in a multicentric design in order to achieve a sufficient statistical power level. The following (in descending order of frequency) anatomical meningioma locations affected the optic nerve and were included in the analysis: Sphenoid wing (n = 34), olfactory groove (n = 15), planum sphenoidale (n = 8), tuberculum sellae (n = 5), spheno-orbital meningioma (n = 5). This information was added to the section “3.5 Clinical implications of MIB-1 index in different anatomical locations”.
“The discussion part is hard to read and understand. Please revise this part and try to improve the structure.”
Author´s response:
We have added paragraphs to section discussion in order to make it more understandable to the readership. Furthermore, we have revised several sentences to enhance the scientific level.
“Minor points:
- 3, l. 113-116: Please revise sentences, as context is hard to understand.
- 4, l. 148: “)” missing
- Table 1 has to be divided into two tables
- Figure 5. Please adapt the colors in the figure, e.g. spinal meningiomas in the same color in A and B”
Author´s response:
We revised the minor points according to the suggestions of the reviewer. Table 1 was divided into Table 1 and 2. Furthermore, orange lines now display spinal meningioma in both Figure 6(A) and Figure 6(B).
Round 2
Reviewer 1 Report
The authors have answered to almost all my former queries concerning this article. I would just propose to change two more parts:
1) To reinforce presentation, the added figure 5 could benefit form the presence of scale bars and from subfigure parts showing higher magnification of the representative pictures.
2) In the following part on lines 95-96: "Patients who underwent a Simpson grade ≥III resection were included because partially or subtotally resected tumors...".
Should the sentence not contain a "not included" or "excluded" ? Else please modify the sentence to make it easier for readers to understand.
Author Response
We thank the reviewer for the constructive suggestions on the improvement of the figure 5 containing neuropathological key findings. We have added bar graphs accordingly and included higher power inserts with the main findings of increased cellularity within meningiomas with substantial infiltrates of macrophages and correspondingly increased proliferation rates. However, we felt that the main take home messages of the figures are somewhat lost when including very high power magnification figures as well as high power magnification of observations of virtual absence of increased proliferation/increased macrophages and, therefore, did not include such picture elements to the figure. We sincerely hope to meet the referee’s concerns with these modifications of the figure.
Furthermore, we revised the sentence in the lines 95-96 regarding the exclusion of partially or subtotally resected meningiomas. The reviewer is absolutely right and we apologize for this mistake. It should have meant exclusion in this sentence. All in all, patients who underwent a Simpson grade ≥III resection were excluded because partially or subtotally resected tumor specimen do not necessarily contain the hotspot area displaying the regions with maximum proliferative activity.